# Conservative Trio-Therapy for Varus Knee Osteoarthritis: A Prospective Case-Study

**DOI:** 10.3390/medicina58040460

**Published:** 2022-03-22

**Authors:** Luise Puls, Dorian Hauke, Carlo Camathias, Thomas Hügle, Alexej Barg, Victor Valderrabano

**Affiliations:** 1Swiss Ortho Center, Schmerzklinik Basel, Hirschgässlein 15, 4010 Basel, Switzerland; luise.puls1@gmail.com (L.P.); dhauke@swissmedical.net (D.H.); 2Praxis Zeppelin, Brauerstrasse 95, 9016 St. Gallen, Switzerland; camathias.carlo@gmail.com; 3Department of Rheumatology, Lausanne University Hospital (CHUV), University of Lausanne, Rue du Bugnon 21, 1011 Lausanne, Switzerland; thomas.hugle@chuv.ch; 4Department of Trauma and Orthopaedic Surgery, University Medical Centre Hamburg-Eppendorf, Martinistrasse 52, 20251 Hamburg, Germany; alexej.barg@googlemail.com

**Keywords:** osteoarthritis, knee, insoles, viscosupplementation, physiotherapy

## Abstract

*Background and Objectives*: Knee osteoarthritis (OA) is a frequent cause of pain, functional limitations, and a common reason for surgical treatment, such as joint replacement. Conservative therapies can reduce pain and improve function; thus, delaying or even preventing surgical intervention. Various individual conservative therapies show benefits, but combination therapies remain underexplored. The aim of this prospective case-study was to assess the effect of a conservative combination therapy in patients with painful varus knee OA. *Materials and Methods*: With strong inclusion and exclusion criteria, nine patients with painful varus knee OA (mean age 56 years (range 51–63 years) were selected and monitored over six months, using the following clinical outcome scores: pain visual analog scale (VAS), Western Ontario and McMaster Universities osteoarthritis index (WOMAC score), short-form–36 items (SF-36) quality of life score, and the sports frequency score. All patients received a standardized conservative trio-therapy with varus-reducing hindfoot shoe-insoles with a lateral hindfoot wedge, oral viscosupplementation, and physiotherapy for six months. *Results*: The pain was reduced significantly from initial VAS values of 5.4 points (range, 3–10) to values of 0.6 points (range, 0–3; *p* < 0.01), at the end of treatment. After six months, seven out of nine patients reported no pain at all (VAS 0). The WOMAC score improved significantly, from initial values of 35 (range, 10–56) to values of 2 (range, 0–9; *p* < 0.01). The SF-36 score showed significant improvement after six months in all four domains of physical health (*p* < 0.01) and in two of the four domains of mental health (*p* < 0.05). The sports frequency score increased by at least one level in six out of nine patients after six months. *Conclusions*: The conservative trio-therapy in patients with varus knee OA showed positive initial clinical results: less pain, higher function, better quality of life, and higher sport activity. Further studies are required to evaluate the long-term effect.

## 1. Introduction

Osteoarthritis (OA) of the knee joint is a progressive degenerative joint disease affecting 10 to 15% in the age group over 60 years [1], with more than 250 million affected people worldwide [2]. Symptoms include pain, stiffness, joint function loss, muscle atrophy, swelling, and reduced mobility [3]. The main risk factors are age, female gender, genetic predisposition, anatomical anomalies, body mass index above 25 kg/m^2^, injuries, and heavy physical work [4,5].

Kellgren and Lawrence distinguished four stages [6]: grade 0 (none), definite absence of X-ray changes of OA; grade 1 (doubtful), doubtful joint space narrowing and possible osteophytic lipping; grade 2 (minimal), definite osteophytes, and potential joint space narrowing; grade 3 (moderate), moderate multiple osteophytes, definite narrowing of joint space and some sclerosis, and possible deformity of bone ends; grade 4 (severe), large osteophytes, marked narrowing of joint space, severe sclerosis, and definite deformity of bone ends [6]. Conservative therapies are beneficial for patients in grade 1 to 3, while grade 4 usually requires surgical intervention [7].

Several studies have shown that physiotherapy mitigates the symptoms of knee OA [8]. Exercises increasing strength and flexibility are most effective. These positive effects are maintained for two to six months after completion of formal treatment [9].

Although it remains controversial, orally administered components of the extracellular matrix (viscosupplementation), in particular glucosamine sulfate and chondroitin sulfate, might relieve pain in joints of lower extremities, including knee [10,11,12,13]. Positive effects of viscosupplementation on stiffness and functionality have also been described [14,15].

In case of varus deviation of the leg axes, shoe insoles with lateral hindfoot wedges can reduce the adduction moment, thereby relieving the medial compartment [16]. A meta-analysis of several studies estimated a pain reduction of 1 point on a scale of 0 to 10, while flat insoles yielded a pain reduction of 0.3 points [17]. In general, insoles are comfortable to wear in shoes and have high compliance. An alternative for this is a soft knee brace. This can align the leg axis and, thus, relieve the medial compartment of the knee [18,19].

There are plenty of other conservative therapies. While some, such as dry needling, have been shown to be not effective [20], others, such as manual therapy, intra-articular viscosupplementation with hyaluronic acid, or platelet rich plasma treatment, have been reported to be effective [21,22,23]. In addition, the gut microbiome could have an impact on the inflammation and pain of OA, but this is still poorly understood [24].

Most previous studies described the effect of individual forms of therapy, but combinations remain underexplored. In this study, patients with varus knee OA were treated with a conservative trio-therapy, a combination of varus-reducing hindfoot shoe-insoles, oral viscosupplementation, and physiotherapy. We hypothesized that this conservative trio-therapy would reduce pain and improve measured scores in patients with varus knee OA.

## 2. Materials and Methods

### 2.1. Patient Recruitment

In this prospective case-study, all patients received the same trio-therapy and were followed for 6 months. Patients were recruited through referrals from general practitioners over a seven-month period. The inclusion criteria were unilateral or bilateral unicompartmental isolated medial varus knee OA with varus > 5° in orthoradiogram/lower limb radiography and Kellgren-Lawrence Grade II to III OA with-out any signs of lateral knee OA or patellofemoral OA in the MRI, age between 18 and 80 years, willingness to participate in the study, and signed patient’s informed consent. The exclusion criteria were unicompartmental medial varus knee OA with Kellgren-Lawrence Grade I or IV, varus knee OA with lateral or patellofemoral OA, valgus knee OA, patellofemoral OA, complete tri-compartmental knee OA, other diseases of the knee joint (e.g., lesions of the anterior or posterior cruciate ligament, lesions of the medial or lateral meniscus, collateral ligament lesions, arthritis, etc.), age < 18 years and >80 years, hip OA, OA of the ankle joint or foot, other leg or spinal disorders, neuromuscular diseases, rheumatoid diseases, osteoporosis, obesity with BMI > 30 kg/m^2^, and lack of willingness to perform the conservative trio-therapy. With these strict inclusion and exclusion criteria, 80% of the patients had to be excluded from the study. For example, these common combinations had to be excluded: medial degeneration Grade IV, or medial degeneration with medial meniscus degeneration/tear, medial degeneration with patellofemoral OA, and others. In total 12 ‘absolute clean’ patients could be recruited.

### 2.2. Conservative Trio-Therapy

All patients received a standardized trio-therapy with corrective shoe-insoles, oral viscosupplementation, and physiotherapy for six months.

The orthopedic insoles were equipped with a varus-reducing lateral hindfoot wedge to reduce the leg’s varus, i.e., the knee axis’s lateralizing position; thus, relieving the knee OA medially. For each degree in varus, a correction of 1 mm wedge height was implemented in the lateral orthopedic insole. The insoles were all performed by the same orthopedic technician and fitted in flat mid-height shoes with hard soles. The patients were instructed to wear the shoe throughout the entire day.

Oral viscosupplementation: patients ingested one sachet (21.8 g) of ExtraCellMatrix ECM^®^ Drink (Swiss Alp Health, Belmont-sur-Lausanne, Switzerland) once daily for the first three months. According to Swiss Alp Health recommendations, maintenance dosing during months four to six was two to three sachets per week. The ExtraCellMatrix ECM^®^ Drink consisted of the following ingredients: collagen type I, II, III (10 g), glucosamine sulfate (1270 mg), chondroitin sulfate (500 mg), hyaluronic acid (100 mg), MSM (100 mg), L-lysine (400 mg), L-threonine (300 mg), rosehip, melon extract, edelweiss, gen-tian, agar–agar, paprika extract, minerals (Ca, Mn, Cu, Zn, Se), and vitamins (C, E, D3, K2).

All patients received physiotherapy twice a week for 30 min, over the entire period of six months. Uniform stretching hamstrings, quadriceps calf muscles, strength training of all periarticular knee muscles, and local anti-inflammatory measures to reduce pain and improve functionality, e.g., local lymphatic drainage, were prescribed. The exercises were adjusted to the patient, according to their pain tolerance.

### 2.3. Scoring

Patients were interviewed at the start of therapy and after three and six months. The level of pain was assessed with the visual analog scale (VAS) on a scale of 0 (no pain) to 10 (most severe pain) [25]. Functional disability was calculated using the Western Ontario and McMaster Universities Osteoarthritis Index (WOMAC-Score) [26]. This disease-specific questionnaire measures pain in specific situations, stiffness, and physical difficulty in everyday activities [26]. The number of points achieved was converted into a percentage (%) of physical functional impairment. The average population aged 45 to 64 years achieves a functional impairment of 8% on a scale from 0 to 100%. A change of at least 10% is clinically relevant [27].

Quality of life was assessed using the short form–36 items (SF-36) questionnaire, before starting therapy and after six months [27]. This questionnaire records physical and mental health in eight domains. The four domains of physical health are physical functioning (PF), role-physical (RP), bodily pain (BP), and general health (GH). The four domains of mental health are vitality (VT), social functioning (SF), role-emotional (RE), and mental health (MH). The highest possible score per domain is 100 points, on a scale from 0 to 100 points and is achieved by healthy people without limitations or disabilities. The parts general health, vitality, and mental health are ‘bipolar’ and have a broader range in their coverage of negative and positive health states. Here, 50% is achieved with a neutral assessment (neither positive nor negative). One hundred points are achieved in these domains if the patients describe their state of health as positive [28]. The values of the US-American average population served for comparison with the study results [28].

Physical sports activity was assessed with the sports frequency score (Table 1) on all three dates [29].

### 2.4. Statistical Analysis

A power calculation before the study started resulted in a minimum number of four patients for a VAS pain score (scale from 0 to 10 points) with the following average values: before intervention OA pain VAS 6, after trio-therapy VAS 2, standard deviation 2, statistical significance *p* < 0.05, statistical power 0.8. A chi-square likely ratio test was used to assess the statistical significance of differences between the start and end of treatment. The significance level was set at α = 0.05. The results are depicted as average values with ± standard deviations and the range of minimum and maximum values. We also calculated minimal detectable changes (MDC) for a power of 0.8.

## 3. Results

Of the 12 patients initially enrolled in the study, nine patients could be thoroughly examined and interviewed till the endpoint of six months follow-up. Two patients could not be reached after the first appointment and were, therefore, excluded from the data analysis. Another patient declined the conservative therapy after the first appointment and had surgery on the affected knee. The data of the three drop-out-patients were not used. Therefore, nine patients could be used in the study: seven women and two men, with a mean age of 56 years (range, 51–63 years) and a mean varus deviation of 7° (range, 5–10°).

### 3.1. Pain Level

All patients reported a reduction of pain in the affected knee (Figure 1 and Figure 2). At the start of therapy, patients reported pain levels of VAS 5.4 ± 2.3 (range, 3–10). After three months of trio-therapy, patients reported pain levels of VAS 1.7 ± 2 (range, 0–5), *p* < 0.01. After six months of therapy, patients reported pain levels closed to the minimum value possible: VAS 0.6 ± 1.1 (range, 0–3) *p* < 0.01. Seven out of nine patients reported no pain at all (VAS 0) after six months. Most of the improvement was achieved within the first three months. The MDC of 2.5 was reached.

### 3.2. WOMAC Score

Patients had initial WOMAC scores of 35 ± 15 (range, 10–56) (Figure 3 and Figure 4). During therapy, values declined to 4 ± 5 (range, 0–14; *p* < 0.05) after three months of therapy, and to 2 ± 4 (range, 0–9; *p* < 0.01) after six months. Thus, the patients achieved better scores than the normal population in this age category [27]. Five out of nine patients achieved the ideal value of 0. Most of the improvement was achieved within the first three months. The MDC of 16.2 was reached.

### 3.3. Quality of Life

The SF-36 questionnaire assessed quality of life before the start of therapy and after six months (Figure 5). All four domains of physical health (physical functioning, role-function, bodily pain, general health) showed significant improvements (*p* < 0.01). In contrast to the VAS results, the SF-36 showed complete freedom from pain in only two patients. It is important to note that the SF-36 questionnaire refers to the entire body and mental state, while in this study, the VAS only referred to the knee. It is, therefore, possible that the patients had other present problems besides the knee pain.

In the four domains of mental health, the two domains, vitality and mental health showed significant improvements (*p* < 0.05). The two domains role-emotional and social functioning showed only minor non-significant changes (*p* > 0.05). The two domains bodily pain and general health did reach the MDC, the other six domains didn’t. 

### 3.4. Sports Frequency Score

Sports physical activity increased by at least 1 level on the sports frequency score in four out of nine patients within the first three months, and six out of nine patients after six months (Figure 6). Running was most popular, followed by fitness, skiing, cycling, and tennis.

## 4. Discussion

Knee osteoarthritis (OA) is a common disease in the middle to old age. Patients with a substantial leg axis deviation (varus/valgus) are frequently affected. Surgery or joint re-placement is usually necessary in the long term. Conservative therapies can relieve pain and improve function, at least temporarily and especially in the early stages. Although various conservative treatments are effective, the possibly enhanced benefits of combinatorial therapies are underexplored. In this prospective study, it was shown that a six-month conservative trio-therapy consisting of varus-reducing hindfoot shoe-insoles with a lateral wedge, oral viscosupplementation with ECM, and physiotherapy enormously improved pain, function, quality of life, and sports activity in patients with painful varus knee OA.

The conservative trio-therapy achieved an average pain relief of 5 points (scale 0 to 10 points) and an improved WOMAC functionality of 30 points (scale 0 to 100 points). Seven out of nine patients even became entirely pain-free in the affected joint within six months. In six of the eight domains of the SF-36 questionnaire, there was a significant improvement in life quality. Physical activity also increased in six of the nine patients by at least one level on the sports frequency score.

The success of the combined trio-therapy in this study was striking compared to previous studies using monotherapies. A meta-analysis of studies on physiotherapy alone showed a pain reduction of only 1 point on a scale of 0 to 10 points and improved functionality of only 10 points on a scale of 0 to 100 points [9]. Insoles alone with lateral side wedges resulted in a pain reduction of 1 point on a scale of 0 to 10 points in previous studies [17]. Viscosupplementation with chondroitin sulfate alone resulted in a pain reduction of 1 point on a scale of 0 to 10 points compared to the control group [10]. Surprisingly, the administration of chondroitin sulfate, together with glucosamine sulfate, had an even worse effect than placebo (pain reduction of only 1 point compared to 2 points with placebo) [12]. However, the combination of physiotherapy and oral viscosupplementation (glucosamine hydrochloride, chondroitin sulfate, bio-curcumin) reduced pain by 2–3 points on VAS and improved function by about 15 points (WOMAC score) [30]. Nevertheless, previous studies on oral viscosupplementation had methodological shortcomings because OA involving different axis deviations were pooled instead of being analyzed as varus/valgus/neutral separately. Moreover, a prerequisite for the effect of oral viscosupplementation could be that biomechanics should be improved simultaneously using corrective insoles and physiotherapy.

Assuming purely additive effects of the individual components, a pain reduction of 4 points could be expected for the conservative trio-therapy, based on the study data to date. This value is close to the observed results of the present study, of 5 points. Additive effects are compatible with the targeting of different pathophysiological aspects. Overall, the pain reduction and improved function were highly effective, suggesting this conservative combination therapy as the first choice treatment in varus knee OA.

This study has limitations. The patient population of nine patients was small. Nevertheless, according to the power calculation, this was more than sufficient to answer the question. Furthermore, there was no control group in this prospective case-study, so that part of the improvement could have been based on the placebo effect. Therefore, it is not known how much of the changes can be attributed to the three forms of treatment together. Double-blinding is not possible for physiotherapy and insoles, so that the placebo effect cannot be determined precisely. Observations with flat insoles without side wedges suggest only a minimal placebo effect for insoles with lateral side wedges [17]. A possible placebo effect for oral viscosupplementation was demonstrated, in the range of a maximal 2 points (pain scale from 0 to 10) [12].

Furthermore, the study had a duration of only six months. Therefore, it is unclear how long the pain reduction lasts and how long a surgical intervention can be delayed compared to patients without a conservative trio-therapy. These questions should be clarified in future studies.

## 5. Conclusions

The conservative trio-therapy with varus-reducing hindfoot shoe-insoles with a lateral wedge, oral viscosupplementation with ECM, and physiotherapy in patients with varus knee OA showed positive initial clinical results: less pain, higher function, better quality of life, and higher sport activity. Further studies are required to evaluate the long-term effect. If this trio-therapy should become established in daily clinical practice, it would provide a treatment option for most patients with varus knee OA.

## Figures and Tables

**Figure 1 medicina-58-00460-f001:**
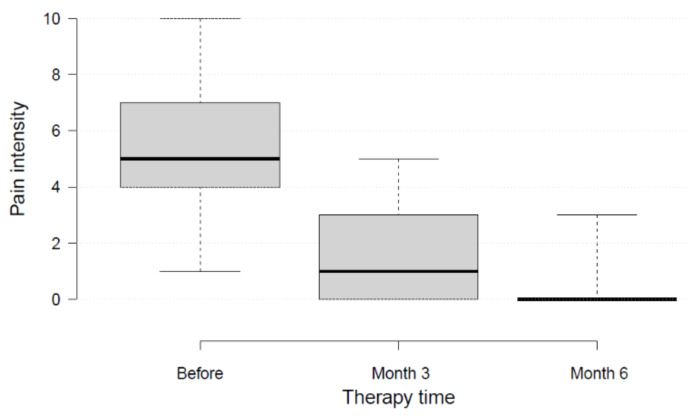
Pain intensity before therapy, and after three and six months of treatment.

**Figure 2 medicina-58-00460-f002:**
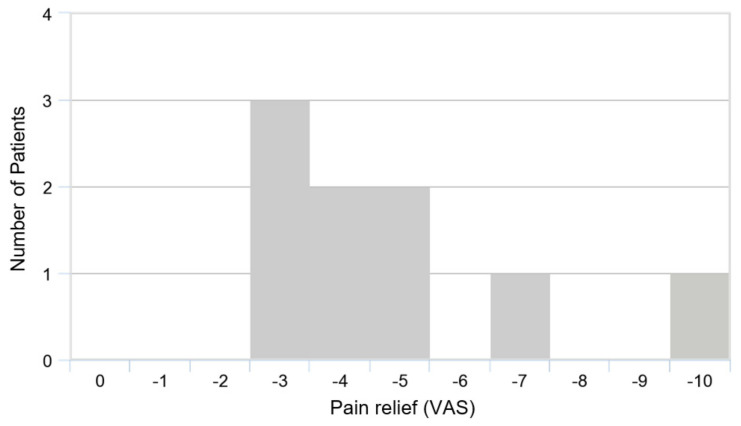
Pain relief in total, over a period of six months.

**Figure 3 medicina-58-00460-f003:**
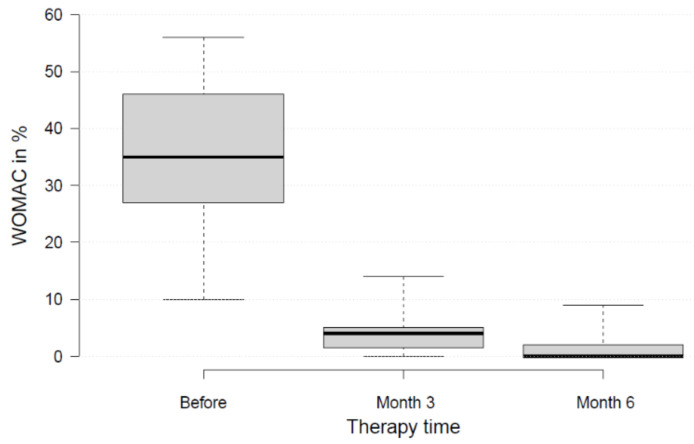
WOMAC score before therapy, and after three and six months of treatment.

**Figure 4 medicina-58-00460-f004:**
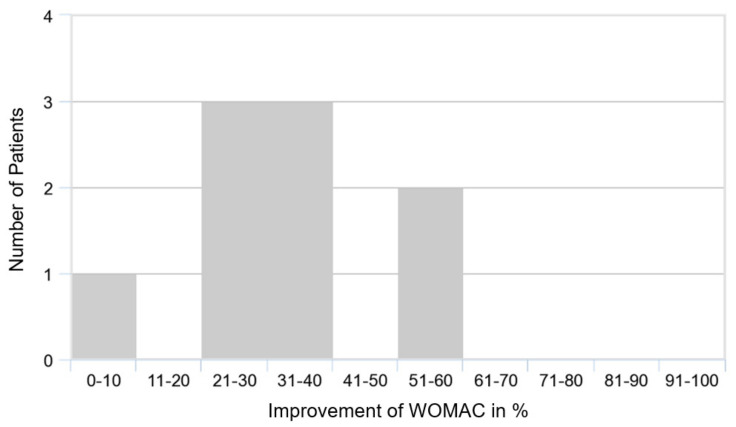
Improvement of the WOMAC score in total, over a period of six months.

**Figure 5 medicina-58-00460-f005:**
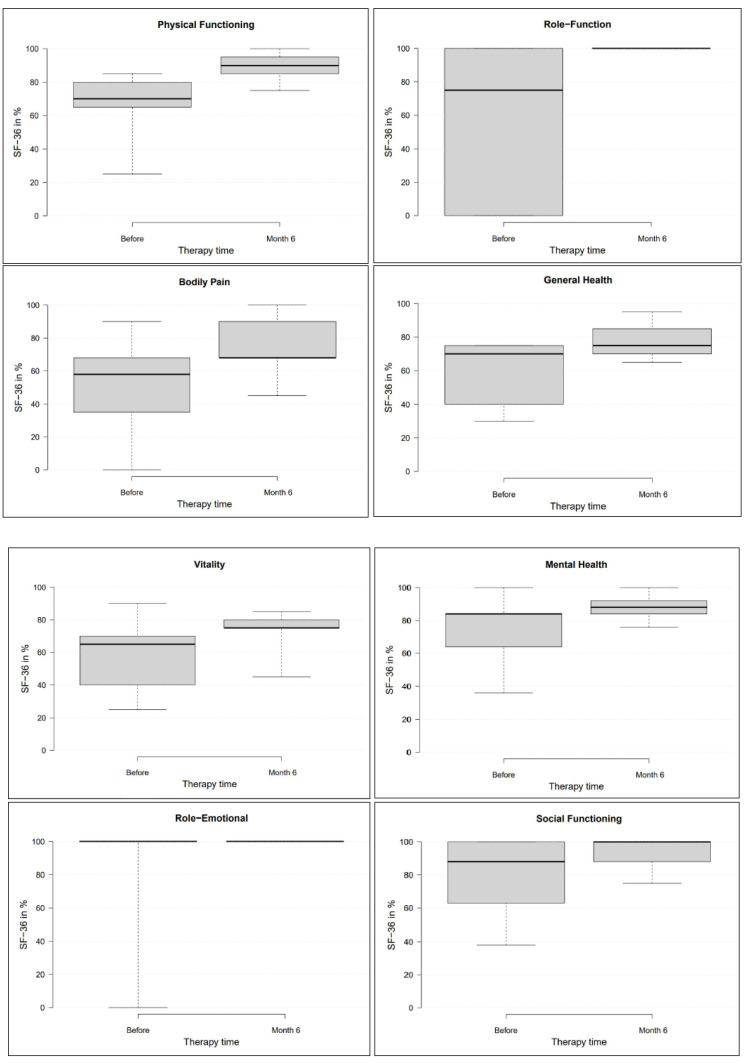
The eight domains of the SF-36 questionnaire before therapy and after six months.

**Figure 6 medicina-58-00460-f006:**
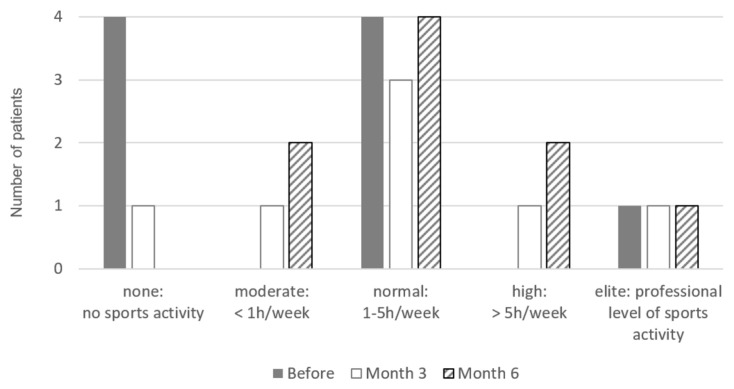
Sports frequency score before therapy, and after three and six months of treatment.

**Table 1 medicina-58-00460-t001:** Sports Frequency Score [29].

Score	Definition
0 (none) 1 (moderate) 2 (normal) 3 (high) 4 (elite)	No sports activity Moderate level of sports activity in leisure time, <1 h/wk. Normal level of sports activity in leisure time, 1–5 h/wk. High level of sports activity in leisure time, >5 h/wk. Professional level of sports activity, elite athlete.

## Data Availability

The data presented in this study are available on request from the corresponding author. The data are not publicly available, due to privacy and ethics.

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
