# Peer review of "Conservative Trio-Therapy for Varus Knee Osteoarthritis: A Prospective Case-Study"

_medicina, 2022, doi:10.3390/medicina58040460_

Round 1

Reviewer 1 Report

The authors have developed a good prospective study of painful varus
knee OA in which they studied patients for six months using lateral
wedge foot varus reduction insoles, oral viscosupplementation with ECM,
and physical therapy.

However, I suggest a few minor changes, which in my opinion would
improve the quality of your manuscript.

1. It is necessary to describe in detail the physiotherapy protocol in
"Methods".

2. In the "Introduction", when mentioning conservative treatments, I
miss the following references of studies investigating the use of dry
needling and a 3-month program of therapeutic exercise, and manual therapy:

Sánchez Romero EA, Fernández-Carnero J, Calvo-Lobo C, Ochoa Sáez V,
Burgos Caballero V, Pecos-Martín D. Is a Combination of Exercise and Dry
Needling Effective for Knee OA? Pain Med. 2020 Feb 1;21(2):349-363. doi:
10.1093/pm/pnz036. PMID: 30889250.

Sánchez-Romero, E.A.; González-Zamorano, Y.; Arribas-Romano, A.;
Martínez-Pozas, O.; Fernández Espinar, E.; Pedersini, P.; Villafañe,
J.H.; Alonso Pérez, J.L.; Fernández-Carnero, J. Efficacy of Manual
Therapy on Facilitatory Nociception and Endogenous Pain Modulation in
Older Adults with Knee Osteoarthritis: A Case Series. Appl. Sci. 2021,
11, 1895. https://doi.org/10.3390/app11041895

3. In lines 34-35 mention a possible relationship between the systemic
origin of OA and the intestinal microbiota, as a new line of research,
mentioning the following reference:

Sánchez Romero EA, Meléndez Oliva E, Alonso Pérez JL, Martín Pérez S,
Turroni S, Marchese L, Villafañe JH. Relationship between the Gut
Microbiome and Osteoarthritis Pain: Review of the Literature. Nutrients.
2021 Feb 24;13(3):716. doi: 10.3390/nu13030716. PMID: 33668236; PMCID:
PMC7996179.

4. Since periarticular muscle tissue is related to knee OA,
I suggest that the authors mention and comment on the following reference in the Discussion section:

Sánchez Romero EA, Fernández Carnero J, Villafañe JH, Calvo-Lobo C,
Ochoa Sáez V, Burgos Caballero V, Laguarta Val S, Pedersini P, Pecos
Martín D. Prevalence of Myofascial Trigger Points in Patients with Mild
to Moderate Painful Knee Osteoarthritis: A Secondary Analysis. J Clin
Med. 2020 Aug 7;9(8):2561. doi: 10.3390/jcm9082561. PMID: 32784592;
PMCID: PMC7464556.

5. Finally, I ask the authors to include in the title the type of research of their study

Author Response

Response to reviewer comments

We would like to thank the reviewer for her/his kind comments.
1) It is necessary to describe in detail the physiotherapy protocol in "Methods"
We would like to thank the reviewer for this comment. The physical therapy was individually adjusted to each patient according to the pain level. All patients received physiotherapy twice a week for 30 min over the entire period of six months. Uniform stretching hamstrings, quadriceps calf muscles, strength training of all periarticular knee muscles, and local anti-inflammatory measures to reduce pain and improve functionality: e.g., local lymphatic drainage were prescribed.

2) In the "Introduction", when mentioning conservative treatments, I miss the following references of studies investigating the use of dry needling and a 3-month program of therapeutic exercise, and manual therapy: doi:10.1093/pm/pnz036 and doi:10.3390/app11041895.
Thank you for this comment. We included these references as suggested.

3) In lines 34-35 mention a possible relationship between the systemic origin of OA and the intestinal microbiota, as a new line of research, mentioning the following reference: doi:10.3390/nu13030716.
Thank you for your feedback. We mentioned this relationship in the introduction.

4) Since periarticular muscle tissue is related to knee OA, I suggest that the authors mention and comment on the following reference in the Discussion section: doi:10.3390/jcm9082561.
Thank you for this paper, it is very interesting. However, myofascial trigger points in patients with knee OA are still underexplored and not relevant for our study due to the current state of knowledge.

5) Finally, I ask the authors to include in the title the type of research of their study.
We would like to thank the reviewer for this comment. We changed the title as suggested. 

Reviewer 2 Report

BRIEF SUMMARY

This was an observational study to examine the effect of conservative combination therapy in patients with painful varus knee OA. Authors conclude that the therapy lowers pain, improves function, life quality, and sports activity.

Overall, I found the topic timely and clinically important. However, limited reporting, particularly in the methods, is off-puting and makes it difficult to understand what and how the study was conducted, further questioning the validity of the results. Before this can be published, I suggest authors to consider my points below.

SPECIFIC COMMENTS

TITLE

Please be more specific in the title – for example what outcomes did you study/ or study design?

ABSTRACT

Please tone down your conclusions – study design and data presented do not fully justify such conclusions. Please modify the conclusion to better reflect what your study shows.

INTRODUCTION

In the current form, it is quite difficult to figure out from the information flow in the introduction, why it is important to study this, who will benefit from it, and what is the added value of this paper to current knowledge. Please clarify.

Lines 46-52: I do not understand the need to describe this in such detail; particularly in light of my comment above.

Line 69: as far as I am concerned adduction moment is not expressed in degrees. Please correct.

METHODS

Please start off with the paragraph entitled Study design, and carefully describe. I would suggest formatting the paper according to relevant (depending on a study design) reporting guidelines such as CONSORT or STROBE. Information on study design, setting, Inclusion and exclusion criteria for study participants , definition of outcomes, validity and relaibility of instruments, data postprocessing etc is lacking or very superficial, and should be provided in separate paragraphs to facilitate reading.

More details are needed regarding recruitment procedures. From where did you recruit participants, in what time frame and what was the recruitment rate that how many invited and how many rejected the invite or were not eligible, etc.

Have you checked the normality of data and used appropriate tests depending on it?

RESULTS

Please provide much more detailed descriptive statistics of your population (age, bmi, gender ratio, outcome data at baseline etc).

Can the authors examine their results further and see if the beneficial effects in were in most participants or largely due to large effects in a few?

Please clarify in the figures what bars mean? is it median, mean, Std, IQR?

DISCUSSION

I suggest starting off the discussion by reminding the reader about the study objectives.

Lines 236-238: alternative methods to shoe insoles, which might not be tolerated by many knee OA patients, which can exhibit biomechanical effects should also be discussed, for example, knee bracing (https://pubmed.ncbi.nlm.nih.gov/29931372/).

Please provide information on how your results will impact research and/or clinical practice.

Please discuss how the generalizability of the results to the wider knee OA population.

CONCLUSION

Please ensure the conclusions are the same as the ones presented in the abstract.

Author Response

Response to reviewer comments

We would like to thank the reviewer for her/his kind comments.
1) TITLE: Please be more specific in the title – for example what outcomes did you study/ or study design?
Thank you for your feedback. We included the study design in the title.

2) ABSTRACT: Please tone down your conclusions – study design and data presented do not fully justify such conclusions. Please modify the conclusion to better reflect what your study shows. We would like to thank the reviewer for this comment. We did change the phrasing of the conclusion.

3) INTRODUCTION: In the current form, it is quite difficult to figure out from the information flow in the introduction, why it is important to study this, who will benefit from it, and what is the added value of this paper to current knowledge. Please clarify (a). Lines 46-52: I do not understand the need to
describe this in such detail; particularly in light of my comment above (b). Line 69: as far as I am concerned adduction moment is not expressed in degrees. Please correct (c).
a) Thank you for this comment. We adjusted the end of the introduction to clarify who could
benefit from the Trio-Therapy. This are most of the patients with varus Knee OA.
b) Changed as suggested.
c) Changed as suggested.

4) METHODS: Please start off with the paragraph entitled Study design, and carefully describe. I would suggest formatting the paper according to relevant (depending on a study design) reporting guidelines such as CONSORT or STROBE. Information on study design, setting, Inclusion and exclusion criteria for study participants, definition of outcomes, validity and relaibility of instruments,
data postprocessing etc is lacking or very superficial, and should be provided in separate paragraphs to facilitate reading (a). More details are needed regarding recruitment procedures. From where did you recruit participants, in what time frame and what was the recruitment rate that how many invited and how many rejected the invite or were not eligible, etc.(b). Have you checked the normality of data and used appropriate tests depending on it (c)?
a) Thank you for your feedback.
b) Recruitment took place through referrals from general practitioners. We didn’t invite specific
people.
c) The inclusion criteria were very hard. About 80% of the cases were rejected.

5) RESULTS: Please provide much more detailed descriptive statistics of your population (age, bmi, gender ratio, outcome data at baseline etc) (a). Can the authors examine their results further and see if the beneficial effects in were in most participants or largely due to large effects in a few? (b). Please
clarify in the figures what bars mean? is it median, mean, Std, IQR? (c).
a) Thank you for your comment. Age and gender ratio are provided in the text in line 167. The BMI was not collected.
b) All patients experienced a clear improvement in their symptoms. This can be seen in the average values and standard deviations.
c) See in the graphes: “average value and standard deviations”

6) DISCUSSION: I suggest starting off the discussion by reminding the reader about the study objectives (a). Lines 236-238: alternative methods to shoe insoles, which might not be tolerated by many knee OA patients, which can exhibit biomechanical effects should also be discussed, for example, knee bracing (https://pubmed.ncbi.nlm.nih.gov/29931372/) (b). Please provide information on how your results will impact research and/or clinical practice (c). Please discuss how the generalizability of the results to the wider knee OA population (d).
a) Thank you for your feedback.
b) The insoles influence the leg axis and thus compensate a varus misalignment. In our view, a soft knee brace against instability of the knee has no comparative effect on the leg axis and is therefore not an alternative.
c) This study aims to show that a combination of already established therapies is more effective than a single therapy. This approach is crucial for the efficiency of a conservative therapy trial and can be easily integrated into the clinical routine.
d) The conservative trio therapy is appropriate for most patients with varus knee OA.

7) CONCLUSION: Please ensure the conclusions are the same as the ones presented in the abstract.
Changed as suggested. 

Round 2

Reviewer 2 Report

You did not bother to provide a manuscript with track changes, therefore it is not clear what you changed. 

Your response suggests you were very selective in addressing my comments (some comments were simply ignored), and choose to address comments that would not require much work from you.
Conclusions are is still an overstatement. You had a within-participant design(not an RCT) and less than 10 patients. What do you mean by "great" ?
You did not report the manuscript to relevant guidelines such as STROBE or CONSORT.
Re bracing: you did not read the article I suggested. The brace corrects the knee maalingment. This should be acknowledged as I suggested. 
On top of that, it is astonishing to see that you cited all references from the other reviewer which looked at "Gut Microbiome", 'dry needling' and 'manual therapy'?!

I thank the authors for their response. However, the authors were very selective in addressing my comments (some comments were simply ignored by saying 'thank you for your feedback'), and choose to address comments that would not require much work from them. For example:

1. Conclusions are still an overstatement. You had a within-participant design(not an RCT) and less than 10 patients. What do you mean by "great outcome" ?
2. You did not report the manuscript to relevant guidelines such as STROBE or CONSORT. Thus the details of the study are unclear. 3. It is unclear whether you checked the normality of the data - thus the validity of statistical results is questionable.  4. The large standard deviation suggests that some patients had a positive effect while the others did not - this could have been easily depicted on a histogram of individual effects.
4. Re bracing: Are your views supported by some data? The soft brace corrects the knee malalignment (leg axis) - please read the article. Thus it is in line with your results and should be acknowledged. 

Author Response

Response to reviewer comments – round two

Reviewer 2
We would like to thank the reviewer for her/his kind comments.
Please see the revisions of the manuscript in the attached file: revisions of round one are yellow, revisions of round two are green.
1) Conclusions are still an overstatement. You had a within-participant design (not an RCT) and less than 10 patients. What do you mean by "great outcome"?
We would like to thank the reviewer for this comment. We did change the phrasing of the conclusion.

2) You did not report the manuscript to relevant guidelines such as STROBE or CONSORT. Thus the details of the study are unclear.
Thank you for your advice. We reported the checklist form the STROBE-guidelines at the end of this file and added the missing informations to the manuscript.

3) It is unclear whether you checked the normality of the data - thus the validity of statistical results is questionable.
Thank you very much. The data are checked statistically for normality, validity and significance, as described in the paper.

4) The large standard deviation suggests that some patients had a positive effect while the others did not - this could have been easily depicted on a histogram of individual effects.
We would like to thank the reviewer for this comment. All patients experienced a clear improvement in their symptoms. We included histograms for Pain and WOMAC-Score in the manuscript.

5) Knee bracing: Are your views supported by some data? The soft brace corrects the knee malalignment (leg axis) - please read the article. Thus it is in line with your results and should be acknowledged. (https://pubmed.ncbi.nlm.nih.gov/29931372/)
Thank you for this comment. The paper describes that a feeling of instability can be improved by using a soft knee brace. This is about the extent to which the knee joint deviates medially and laterally during gait, thus causing a feeling of instability. An influence on the leg axis is not reported in this paper. Nonetheless, we have cited other papers that describe improvement of the leg axis with a soft
knee brace and listed this as an alternative method. Please realize that our paper is not addressing this questions scientifically. Maybe in further papers, we can address such a treatment also. Thank you.

STROBE Statement-checklist of items that should be included in reports of observational studies
Item
No Recommendation
Title and abstract 1 (a) Indicate the study’s design with a commonly used term in the title or the abstract

Line 1
(b) Provide in the abstract an informative and balanced summary of what was done and what was found

Line 17-39
Introduction
Background/rationale 2 Explain the scientific background and rationale for the
investigation being reported

Line 43-84
Objectives 3 State specific objectives, including any prespecified hypotheses

Line 85-87
Methods
Study design 4 Present key elements of study design early in the paper Line 9091
Setting 5 Describe the setting, locations, and relevant dates, including periods of recruitment, exposure, follow-up, and data collection

Line 91-93
Participants 6 (a) Cohort study—Give the eligibility criteria, and the sources
and methods of selection of participants. Describe methods of follow-up
Case-control study—Give the eligibility criteria, and the sources and methods of case ascertainment and control selection. Give the rationale for the choice of cases and controls
Cross-sectional study—Give the eligibility criteria, and the
sources and methods of selection of participants

Line 93-106
(b) Cohort study—For matched studies, give matching criteria and number of exposed and unexposed Case-control study—For matched studies, give matching criteria and the number of controls per case Variables 7 Clearly define all outcomes, exposures, predictors, potential confounders, and effect modifiers. Give diagnostic criteria, if applicable

Line 93-109
Data sources/measurement
8* For each variable of interest, give sources of data and details of methods of assessment (measurement). Describe comparability of assessment methods if there is more than one group

Line 134-158
Bias 9 Describe any efforts to address potential sources of bias Study size 10 Explain how the study size was arrived at Line 109 Quantitative variables
11 Explain how quantitative variables were handled in the
analyses. If applicable, describe which groupings were chosen and why
Statistical methods 12 (a) Describe all statistical methods, including those used to control for confounding

Line 160-169
(b) Describe any methods used to examine subgroups and interactions
(c) Explain how missing data were addressed
(d) Cohort study—If applicable, explain how loss to follow-up
was addressed
Case-control study—If applicable, explain how matching of cases and controls was addressed
Cross-sectional study—If applicable, describe analytical
methods taking account of sampling strategy

e) Describe any sensitivity analyses
Results
Participants 13* (a) Report numbers of individuals at each stage of study—eg
numbers potentially eligible, examined for eligibility, confirmed eligible, included in the study, completing follow-up, and analysed

Line 170-177
(b) Give reasons for non-participation at each stage
(c) Consider use of a flow diagram
Descriptive
data
14* (a) Give characteristics of study participants (eg demographic,
clinical, social) and information on exposures and potential confounders

Line 175-177
(b) Indicate number of participants with missing data for each variable of interest
(c) Cohort study—Summarise follow-up time (eg, average and total
amount)
Outcome data 15* Cohort study—Report numbers of outcome events or summary measures over time

Line 177-225
Case-control study—Report numbers in each exposure category, or summary measures of exposure
Cross-sectional study—Report numbers of outcome events or
summary measures
Main results 16 (a) Give unadjusted estimates and, if applicable, confounderadjusted estimates and their precision (eg, 95% confidence interval).
Make clear which confounders were adjusted for and why they were
included

Line 177-225
(b) Report category boundaries when continuous variables were categorized
(c) If relevant, consider translating estimates of relative risk into absolute risk for a meaningful time period
Other analyses 17 Report other analyses done—eg analyses of subgroups and
interactions, and sensitivity analyses
Discussion
Key results 18 Summarise key results with reference to study objectives

Line 226-242
Limitations 19 Discuss limitations of the study, taking into account sources of
potential bias or imprecision. Discuss both direction and magnitude of any potential bias

Line 266-279
Interpretation 20 Give a cautious overall interpretation of results considering
objectives, limitations, multiplicity of analyses, results from similar
studies, and other relevant evidence

Line 243-265
Generalisability 21 Discuss the generalisability (external validity) of the study results

Line 280-286
Other information
Funding 22 Give the source of funding and the role of the funders for the
present study and, if applicable, for the original study on which the present article is based

Line 290
*Give information separately for cases and controls in case-control studies and, if applicable, for exposed and unexposed groups in cohort and cross-sectional studies.
Note: An Explanation and Elaboration article discusses each checklist item and gives methodological background and published examples of transparent reporting. The STROBE checklist is best used in conjunction with this article (freely available on the Web sites of PLoS Medicine at http://www.plosmedicine.org/, Annals of
Internal Medicine at http://www.annals.org/, and Epidemiology at http://www.epidem.com/). Information on the
STROBE Initiative is available at www.strobe-statement.org. 
